# Provider perspectives on the use of motivational interviewing and problem-solving counseling paired with the point-of-care nucleic acid test for HIV care

Dana L. Atkins[1]*, Lauren Violette[1,2], Lisa Neimann[1], Mary Tanner[3], Karen Hoover[3], Deepa Rao[4,5], Joanne D. Stekler[1,2,4]

1 Department of Medicine, University of Washington, Seattle, Washington, United States of America,
2 Department of Epidemiology, University of Washington, Seattle, Washington, United States of America,
3 Center for Disease Control and Prevention, Atlanta, Georgia, United States of America, 4 Department of Global Health, University of Washington, Seattle, Washington, United States of America, 5 Department of Psychiatry and Behavioral Sciences, University of Washington, Seattle, Washington, United States of America

* dlatkins@uw.edu

**Data Availability Statement:** All relevant data are within the manuscript and its Supporting information files.

## Abstract

### Aims

To evaluate provider perspectives on the use of a point-of-care nucleic acid test (POC NAT) and preferential opinions for motivational interviewing (MI) or problem-solving counseling (PSC) as an ultra-brief intervention for patients experiencing challenges to antiretroviral therapy (ART) adherence.

### Methods

A qualitative study was conducted among providers at an HIV care clinic in Seattle, Washington. Ten in-depth interviews with HIV care providers were completed, which explored determinants of acceptability, feasibility and preferences for a combined adherence counseling and POC NAT intervention for patients living with HIV. Interviews were analyzed through consensus coding and the Five A's Framework to inform thematic analysis.

### Results

Providers favored the use of a combined adherence counseling technique and POC NAT for their non-adherent patients living with HIV. Providers believed the intervention was an improvement on current assessment and advising practices. However, concerns about extended wait times for the POC NAT results influenced perceptions about feasibility around clinic flow and incorporation into clinic practice. Providers believed that acceptability of POC NAT implementation would be enhanced by including a subset of patient populations whom tend to be in the clinic for longer periods, and in tandem face greater ART adherence challenges.

**Funding:** This project received financial support from the Center for Disease Control & Prevention (6 U01 PS 005196-01-00). Two co-authors of the manuscript are employed by the funder (MT and KH), and were involved in the study concept and review of the manuscript. All other authors (DLA, LV, DR, and JDS) received salary support using the grant award paid from the CDC to the University of Washington.

**Competing interests:** The authors have declared that no competing interests exist.

## Conclusion

The GAIN Study will be the first project to evaluate the implementation of POC NAT in the U. S. Continued formative work is ongoing and may illustrate how best to address feasibility and concern around the two-hour time to result. The planned GAIN study will incorporate some of the findings found in this qualitative study and pilot this intervention, including a time-in-motion analyses of clinic flow, which may help reduce perceived wide-scale adaptation of POC NAT and ART adherence counselling among PLHW. Future work, including a shorter time to results and/or lower limit detection could make a significant improvement in the provision of HIV care.

## Introduction

Rigorous adherence to antiretroviral therapy (ART) is essential to avoid the risks of HIV drug-resistance, subsequent progression of disease and increased risk of HIV transmission [1, 2]. Strict medication adherence, however, can be difficult to maintain for many patients and may require both monitoring and support. One strategy for adherence monitoring and support is provider-led counseling.

There are many structured interventions that HIV care providers can use to address ART adherence challenges including: individual counseling, support groups, family-centered services, and treatment supporters [3]. However, one of the most recognized is motivational interviewing (MI)–a person-centered counseling style that seeks to address ambivalence and elicit patient motivation for change. MI has been shown to be effective in improving adherence [4] and other health outcomes among persons living with HIV (PLWH) [5]. An alternative structured counseling intervention is problem-solving counseling (PSC)–a patient-driven discussion to identify and address challenges to sustained health behavior changes [6]. PSC has been used successfully in ART adherence and depression [7].

It is currently unclear to what extent HIV care providers are familiar with these structured counseling interventions, and whether they are employing them in their practices. Some studies have failed to show an impact on ART adherence from MI. More research is warranted to investigate non-motivational challenges as adherence is determined by both motivational and non-motivational factors [8]. However, even with some mixed results on efficaciousness, the overall literature indicates MI can have an important impact on behavior change. PSC has also been shown to have favorable outcomes in co-morbidities—such as addressing depression and ART adherence or initiation in both Zimbabwe [9] and Malawi [10] among adult patients. In addition, a combination of both MI and PSC has been shown to be effective and acceptable in reducing substance use in emergency department patients [11] and young women [12] in South Africa, where both mental health and substance abuse share a direct link to ART adherence challenges. Additionally, two randomized trials are ongoing in South Africa to further illustrate the efficacy of MI and PSC in ART adherence for populations struggling with alcohol abuse and depression [11, 13].

The other critical component of adherence monitoring is HIV nucleic acid (i.e. "viral load") testing (NAT), which can be used as a surrogate marker for HIV clinical outcomes. In some contexts, patient knowledge of viral load has even helped improve medication adherence [14]. However, HIV NAT may not be universally available due to cost and technical issues, and there is a delay in time to results for laboratory-based testing [15]. Point-of-care nucleic

acid tests (POC NAT) have shown promise in expediting the receipt of viral load results and increasing virologic suppression, especially in resource-limited settings [16, 17] and has been used in our previous study, Project DETECT [18, 19].

The GAIN Study is a Centers for Disease Control and Prevention (CDC) funded study that will evaluate the implementation of POC NAT combined with either MI or PSC counseling strategies in a clinic setting and a community setting in Seattle, WA. As part of the formative work for the GAIN Study, this qualitative study aimed to gather medical provider perspectives on the feasibility, acceptability and formative development of a brief POC NAT-based adherence combined with either MI or PSC counseling techniques as an intervention for Madison Clinic providers to use during clinic visits. Through the findings of this qualitative study, we seek to gather further information for successful implementation of the GAIN study and future dissemination of the POC NAT, MI and PSC into real-world clinical practice in HIV clinics in the United States.

## Materials & methods

### Study design and population

Between July and September 2020, this qualitative study recruited medical providers from the Madison clinic, a Ryan White-funded clinic at Harborview Medical Center where the larger GAIN Study that this project was embedded in will enroll. This qualitative work followed the COnsolidated criteria for REporting Qualitative research (COREQ) guidelines (S1 Checklist). We used content analysis design and convenience purposive sampling in order to obtain a diversity of viewpoints from providers with various durations of providing HIV care and diversity in gender and race/ethnicity. We oversampled physicians providing care in the moderate needs ("Mod") clinic within the Madison clinic due to this clinic catering towards patients that oftentimes are experiencing adherence challenges attributed to socioeconomic, mental and addiction difficulties. A total of 11 in-depth interviews (IDI) with HIV care providers were conducted, however only 10 were recorded and therefore included in the analysis (Table 1).

### Ethics statement

The University of Washington Institutional Review Board approved the study. All study participants provided verbal informed consent, and participants were offered a $5 coffee card for their time.

### Data collection and analysis

The IDI discussion guide was developed to understand providers' current adherence counseling practices and their perspectives on implementation of an ultra-brief adherence intervention and the feasibility and acceptability of the implementation of POC NAT into clinical care. Providers were told that MI involves having a safe, nonjudgmental discussion with patients addressing ambivalence to consistent ARV medication use. Providers were told that PSC involves having the patient identify their adherence challenges, listing out possible solutions, and ultimately, helping patients evaluate how achievable each solution is, in order to find the best one.

Providers were recruited by the study Principal Investigator (JDS) via email or phone and asked if they would be interested in participating in an IDI. Interested providers were then scheduled by study staff (LV, LN) for a time that worked best for the provider to conduct the interview. IDIs were conducted by study staff (LV, LN), who are both MPH degree holders

**Table 1. Provider demographics.**

|  | n (%) |
|---|---|
| Age (years)–Median [IQR] | 40 [39, 45] |
| Current gender identity |  |
| Female | 7 (70.0) |
| Male | 3 (30.0) |
| Race/Ethnicity |  |
| White | 7 (70.0) |
| Hispanic or Latino | 1 (10.0) |
| Asian | 1 (10.0) |
| Multiple races | 1 (10.0) |
| Years of clinical HIV care experience |  |
| Median [IQR] | 10 [6. 5, 13] |
| Years of clinical HIV care experience |  |
| 5–7 | 4 (40.0) |
| 8–10 | 3 (30.0) |
| 11–13 | 2 (20.0) |
| 14+ | 1 (10.0) |

*Demographics reflect the 10 provider interviews transcribed, not the 11 interviews completed

and LN has an additional MSW. Both interviewers have prior qualitative data collection experience. IDIs were audio recorded, and transcribed verbatim (DLA, LV, LN). Verbal consent was collected prior to the start of the interview due to the very minimal risk imposed on provider participation in the IDI. Interviews lasted an average of 54 minutes. ATLAS.ti version 8 was used to support data analysis and management. Transcripts were analyzed using directed content analysis guided by an adapted Five A's framework [20]. The Five A's model is based on the trans-theoretical model of behavior [21]; it was originally developed by the US Department of Health and Human Services to encourage smoking cessation [22] and has since been adapted to obesity management [23–25]. For our purposes, we used the Five A's framework to analyze providers' current adherence counseling practices due to its successful use in eliciting behavioral changes in chronic disease management.

Coders (DLA, LV, LN) performed multiple rounds of consensus coding where the same transcript was reviewed and coded by all coders to ensure consistent code application. Following consensus coding, the remaining nine transcripts were divided equally amongst the three coders and individually coded. Transcripts were then traded and coded by another member of the team and disagreements in code application were noted. All disagreements were resolved through group discussion with the larger team. Code co-occurrence tables and queries were used to identify the constructs most influential on intervention utilizations outcomes.

## Results

A total of 11 providers participated in the IDIs; however only 10 interviews were successfully recorded. The majority of the participants were female (73%), white (70%) with a median age of 40 years (IQR: 39, 45), and had been providing HIV care for a median of 10 years (IQR: 6.5, 13). Additional participant demographics are presented in Table 1.

Participants described their current adherence counseling practices and perspectives on the use of a brief MI or PSC technique and POC NAT. Provider responses are shown within the

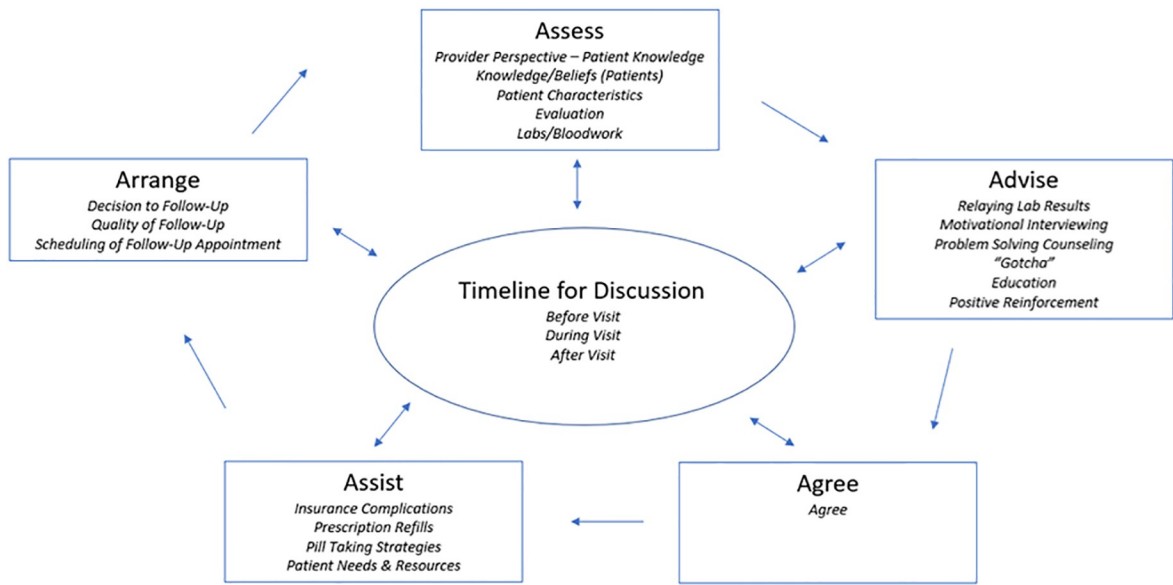

**Fig 1. Adapted 5 A's model of self-management support with thematic in-depth interview code assignments.** Adapted 5 A's Model of self-management support. The codes used in analysis were tied to the main 5 A's model parent codes (Assess, Advise, Agree, Assist and Arrange) with the added component of Timeline for Discussion, which was an entirely new parent code with accompanying child codes.

Five A's framework in Fig 1. The main themes that emerged from the data included: current assessment and advising practices, use if the POC NAT for adherence assessment; Incorporation of the POC NAT into clinic flow; Assessing patient adherence using MI or PSC; Advising patients using MI or PSC; Assisting Patients with Adherence Challenges Using MI and PSC; and Incorporating counseling into clinic practice.

## Current assessment and advising practices

The majority of providers said they begin each appointment by asking open-ended questions, such as *"How are your medications going?"* to elicit patients' most recent experiences taking their ART. Based upon the patient response, providers might then ask targeted questions to probe for further information on ART adherence, asking questions such as *"How many pills have you missed in the last 30 days?"* Although the initial questions tended to be the same between providers, subsequent questions were tailored to the individual patient and often based upon provider knowledge of previous adherence history.

> *"I always check in about meds. I ask them. . . how many doses they think they've missed in the last month. I'll change that, depending on their adherence history. For someone who I know has a harder time or who is not suppressed, I might shorten that window, to try to get a better, more accurate sense from the patient."*
>
> *–Provider, 8 years in HIV care*

Providers often viewed this time with patients as an opportunity to assess issues such as pill fatigue and possible solutions to update current medications to ensure patients continued to be content with their current regimen. Additionally, they identified patient needs and resources that may be causing barriers to adherence, such as prescription refill and insurance

complications. Providers also felt that specific populations may need further assessment and assistance during adherence counseling sessions, especially those experiencing social challenges including unstable housing and drug use and patients where English was a second language.

> *"There are some people who, despite our counseling, they have a lot of issues going on in their lives and HIV is not the most important thing to them or adherence is not something they can comply with given a very stressful psychosocial situation."*
>
> *–Provider, 13 years in HIV care*

To ensure successful adherence counseling, many providers also emphasized the importance of provider-patient relationships. Providers felt that positive, long-standing relationships with patients dramatically influenced how successful adherence counseling conversations were.

> *"[Counseling success] depends on the patient and how I feel our relationship is, as well as the sense of how the patient is interacting with me in that moment. . ..[I]f a patient seems reluctant to talk then I respect that as well. . .[S]ome patients won't say anything [or] they say things that I want to hear or they think I want to hear. [Y]ou have to go along with what the patient feels comfortable with."*
>
> *–Provider, 13 years in HIV care*

Counseling was highly dependent upon when patients completed their blood draws. The majority of patients elected to get viral load monitoring and other laboratory evaluations done after their clinical visit with the physician, thus making adherence counseling based on recent viral loads impossible to do in-person. Providers often are left to relay viral load information via e-care or follow-up phone calls. When choosing the mode of relaying results, providers highlighted the importance of the type of results being given. When results were not as expected, such as when a patient has been historically suppressed but the results showed a detectable viral load, providers wanted to speak with their patients over the phone and not provide this feedback via online messaging. One provider explained,

> *"[F]olks who are virally suppressed [I] often just send them their results over eCare, unless they're requesting a call. . .[B]ut I'd say anybody for whom there's a surprising result, I'm calling them to talk about it."*
>
> *–Provider, 8 years in HIV care*

Providers also highlighted the importance in the quality of their follow-up, often citing how inconsistent it was based on the patient population. They consistently felt that follow-up was particularly difficult with populations of patients who were experiencing socioeconomic hardships such as unstable housing, substance use, English as a second language, and lack of cellphone access. Providers expressed that often the patients who needed follow-up were often the most difficult to get in contact with. As one provider explained,

> *"Some folks are also hard to track down like the folks who are more likely to be not suppressed are the folks who are more likely to maybe not answer their phone if a random or not random number calls them."*
>
> *–Provider, 8 years in HIV care*

These types of challenges illustrate the difficulties in helping patients work through adherence challenges in a holistic, truly patient-centered approach and further emphasize the need for exploring the utilization of such interventions such as MI or PSC with patient populations that experience challenges in adherence to their ART.

## Creation of the GAIN model

Providers were asked about their thoughts on using motivational interviewing and PSC strategies in conjunction with providing POC NAT results during clinic visits. The main findings are provided below, which describe patient needs and challenges to adherence, how the proposed intervention would facilitate assisting patients, and how the intervention could be integrated into the clinic.

## Evaluating the point-of-care NAT intervention using the five A's framework

The first component of the intervention will be the incorporation of POC NAT results into adherence messaging. This will be used to provide real-time semi-quantitative information on a patient's viral load. The Simple Amplification Based Assay version II (SAMBA II) Semi-Q POC NAT has a limit of detection of 1,000 HIV RNA copies/mL (+/- 0.5log) and takes about 2 hours to process [26]. Overall, providers found the idea of incorporating POC NAT results into adherence counseling to be acceptable. However, even with high acceptability, providers speculated that various components would need to be considered to ensure successful implementation.

*"I certainly could see it [the POC NAT] being useful in. . .scenarios where someone reports adherence, but then it comes back, you know, detected . . .[I]t just happened to me last week. So, in that instance, it would have been useful [to have the POC NAT result]."*

*–Provider, 13 years in HIV care*

**Assess: Using the point-of-care NAT for adherence assessment.**   All clinicians supported the use of POC NAT, with many perceiving an advantage of POC NAT over existing testing strategies. Providers emphasized that having a lower limit of detection would be ideal, but that the cut-off of +/- 1000 HIV RNA copies/mL would be acceptable. As one provider stated,

*"I think it'd be nice if we had a lower viral load cut-off, at least like a viral load less than 200. . .[but] there aren't that many people whose viral loads are in the 200 to 1000 range. [A] lot of people are just not taking their medication. So. . . I still think that the viral load cut-off of 1000 will be extraordinarily helpful because you can say for those people that are less than 1000 like, 'Hey, things look to be going well. I'll follow up with you based on your real viral load result.' And then for those that are over 1000, that's very helpful because then we have them still as a relatively captive audience, and can do more adherence counseling."*

*–Provider, 7 years in HIV care*

Clinicians frequently spoke about how the POC NAT may not be useful in assessing all of their patients. One major concern was the 2-hour turn-around time. Another concern providers often spoke about was how the POC NAT intervention would only be beneficial to patients with a history of adherence challenges—particularly those seen in the walk-in "Mod" clinic because they spend more time waiting.

*"I think for most patients. . .that routinely come in for an appointment and leave, that's not going to work. [But thinking of] our walk-in patients who actually do spend time waiting. . .they have other touch points in the clinic, specifically with social work and other services and health educators. . .there's an opportunity to be there [in clinic] for more than two hours. And these are the patients that would benefit most from knowing their results right away."*

*–Provider, 14 years in HIV care*

Providers also felt that the POC NAT results would be beneficial in improving their adherence counseling with patients. By having access to viral load results and not just self-reported adherence, many providers believed that they would have a more frank and open conversation in helping the patient to identify and discuss potential adherence challenges.

*I think. . .mostly it will increase the effectiveness of the counseling. It's very different to have the data right in front of me when the patient is in front of me, and it's not data from two weeks; it's data from now. So, I can tell the patient that you can start changing this today, right? The blood is from today, and the change can start right now. So, I think that's very powerful*

*–Provider, 10 years in HIV care*

Some clinicians also felt the POC NAT was a useful technique to address inaccuracies in patients' adherence self-reports. Others had reservations about its use in this way and emphasized using caution when relaying results to patients. As these two providers explained,

*"[Y]ou could really see if what they're telling you aligns with what the viral load is telling you. . .I think that would just help me to dive more into those barriers. If I don't think what the story they're telling me is consistent with what the viral load is showing."*

*–Provider, 10 years in HIV care*

*"I wouldn't want anybody to think of it as like a, a gotcha kind of test, like, 'oh, you said you were taking your meds and now we have this result. Now we're going to come back and talk about it'. So, I think it needs to be approached in the right way by the provider."*

*–Provider, 6 years in HIV care*

Lastly, providers believed that their patients would consider the POC NAT to be highly acceptable. Many cited that many of their patients would like to be provided with real-time results and the method of blood collection—a fingerstick instead of a venipuncture blood draw —would be appealing. As one provider stated,

*"I think patients would really like getting their viral load result in the visit. I don't know what the trade-off of extra time, like a two-hour wait is, but the fact that they would be able to get it right there, and be able to have that point, especially as we are increasing our emphasis on viral load as this kind of key marker that we're following the patients, I do think that the patients will like that."*

*–Provider, 8 years in HIV care*

Although providers identified advantageous aspects for the use of POC NAT, there were some concerns. Due to messaging around "Undetectable equals Untransmissable" (U = U) which is based on a viral load of <200 copies/mL [27], some providers felt that the POC NAT would not be applicable to these patients or could create confusing messaging.

*"[F]or those people who really want to be undetectable and being able to tell them that 'Yes, you are undetectable today. You can go out and feel good about U = U', because people really want to know that for their sex partners and their confidence and whatnot. So that would not be helpful for the majority of my patients who are doing well with their ART. For the ones who haven't been doing well with their ART, it would be helpful to know at the visit, whether. . .[their] viral load is greater than 1,000.*

*–Provider, 6 years in HIV care*

The incorporation of a test with a two-hour processing time was also a concern addressed by providers. Though many providers had indicated the wait time may be acceptable for a specific population of patients, but it would not be feasible for all.

*"[T]o be honest, I don't know how that would impact clinic flow or how it could be incorporated into clinic flow. I don't think that my patients would wait around for that result."*

*–Provider, 13 years in HIV care*

Others had skepticism on how feasible it would be for appointments scheduled towards the end of the day. As one provider explained:

*[It] seems pretty challenging to do, to be honest. If I'm thinking of. . .develop[ing] a system around this that's going to help. . .get me the result within two hours at a Mod clinic. . .[I]n reality, the way it works is patients check in and if they get seen, by the time I do an H and P. . .that's before two hours' time is up. So, I think there could be a challenge to get [the results] during your interview with the patient. . .[Patients] just show up whenever. So what if you. . . get a bunch of patients around [the end of the day]? You're not going to get those results. [F]rom three o'clock on, those results are not going to be useful in Mod clinic if you want to use those results during your clinic visit. . ."*

*–Provider, 10 years in HIV care*

**Arrange: Incorporating POC NAT into clinic flow.** With the incorporation of the POC NAT into existing clinic flow being the most common concern, providers listed solutions they thought might assist in adoption. For patients with reliable forms of communication, providers suggested calling patients to come in early to complete the fingerstick ahead of the appointment. Others suggested getting the POC NAT started as patients entered the waiting room. As one provider suggested:

*"I think that [the POC NAT] should be done right as a patient. . .get[s] checked in. . .Mod patients. . . usually hang out for almost two hours because they checked in, then they get seen by the nurse and then maybe the social worker comes. . . [to] see them and then they talk with the social worker for a while. . .And then they get some blood drawn so it can almost last two hours. So if the first thing that could happen is that [POC NAT]. . .could get triggered before they see anybody. . .that would be great."*

*–Provider, 10 years in HIV care*

Some providers also discussed how, due to time limitations, utilizing other clinic staff or healthcare professionals to relay results to patients may be a good option. By tying in the whole

healthcare team, one provider expressed how patients could still receive the needed services, counseling and holistic care necessary.

> "[P]harmacists are incredibly valuable and if they have time to sit with a person and discuss issues or review med lists or refill histories or. . . go through an intervention like this, that [would be] extraordinarily useful. So, I think that in instances in which a provider is too pressed for time. . . fully engaging those ancillary team members to help commit the time to do that [would be] incredibly useful."

> –Provider, 13 years in HIV care

## Evaluating motivational interviewing and problem-solving counseling using the five A's framework

**Assess: Assessing patients adherence using MI and PSC.**  Motivational interviewing was viewed by providers as a highly acceptable technique for assessing patient needs and adherence, with all providers stating that they thought the technique would be useful. Four providers said that they felt they already did a version of motivational interviewing in their existing practice. As one provider stated,

> "I love MI. I think it's great if done well. . .I didn't realize I was doing MI, and that's nice to know that [I was]."

> –Provider, 7 years in HIV care

When asked about preference for the use of PSC as a tool of assessing patient needs and adherence, three providers mentioned using this adherence counseling technique. Most were unfamiliar or did not use this technique, and there was no clear preference for its use by any provider interviewed.

**Advise: Advising patients using MI and PSC.**  Providers often felt that in advising patients, education was not often needed. Many openly stated that their patients were educated on the importance of adherence, and therefore it was more pertinent to educate only when there was a specific issue that needed to be addressed. As one provider stated,

> "I tend to avoid long scientific descriptions about why this [not adhering to ART] is bad. Because I don't think in in my practice that people who are having adherence challenges. . .don't understand that their adherence is suboptimal. [T]hey all know that it would be better for them to take their medication. So I try to take a pragmatic approach of saying, 'what are the issues and what can we do to improve them?'"

> –Provider, 7 years in HIV care

**Agree: Patient-provider agreement using MI and PSC.**  Providers often emphasized the importance of patient-provider agreement for successful adherence to be possible. When asked about having patient-provider agreement while using MI, providers often agreed that this would be paramount in its success. As one provider stated,

> "I think having patient buy-in is really important. And getting it to be based on their wants, desires and how that fits in with [their life]."

> –Provider, 14 years in HIV care.

When asked about how agreement would work with using a PSC technique, providers often said they did not use this technique, but they still agreed that patient-provider agreement was essential for its success. Providers felt some of the current adherence challenges were easy issues to find agreement on. As one provider who currently uses PSC explained,

*"I think that my personal approach is a problem-solving approach. . . I mean I have patients who tell me they wanted to stay with [the in-house] clinic pharmacy. . .because they were scared of trying a different location. And I was like, 'Listen, you live so far away. . .this [current pharmacy location] is obviously not working, you pick up your meds 5 to 10 days late every single month. Let's try it at this other pharmacy. I've heard good things about them, and we'll work hard to make sure they have your insurance information. Let's try it and see how it works". [Then I]. . .follow-up with the patient and make sure that they do pick up the medication. I feel like [problem-solving counseling] is a good approach."*

*–Provider, 7 years in HIV care*

However, it was still emphasized by many providers that for both counseling techniques to have patient-provider agreement to improve adherence, underlying challenges must also be addressed. Issues such as mental health issues, addiction and homelessness may supersede ART adherence among a patient's priorities, making improving ART adherence especially difficult to achieve for patients facing these additional challenges.

**Assist: Assisting patients with adherence challenges using MI and PSC.** Overall, providers felt that motivational interviewing would be useful in assisting with their patients' adherence. Most providers felt that their patients wanted to be successful in becoming more adherent and that it really was more about helping patients identify barriers and assisting them in addressing those challenges than generating patient motivation. Providers also believed that going through the process of motivational interviewing with their patients would help them maintain a sense of autonomy and control over their own health. These factors led providers to believe the motivational interviewing was a more empathetic approach to adherence counseling and was therefore highly feasible and acceptable. As one provider said,

*"I think it also just builds rapport with patients; it builds trust with patients. . .[Motivational interviewing can] show that you can be on the same level with patients and to hear from them and let them make their [own] decisions . . .helps create a trusting relationship between patients [and providers]. . .in the clinic."*

*–Provider, 10 years in HIV care*

Though providers felt there was a high relative advantage to the use of motivational interviewing, a few concerns were highlighted during interviews. Some providers felt that learning a new counseling style could be intimidating to other providers, and consequently the intervention may be met with some resistance. As one provider stated,

*"I think the con of [motivational interviewing] is that it sounds intimidating for providers. . .I have been intimidated by motivational interviewing and I'm not exactly sure how to do it. . .but I think having the language [to use] would be really helpful and would mitigate that con."*

*–Provider, 7 years in HIV care*

Providers also highlighted that the success of motivational interviewing would be highly dependent upon by the type of patient being counseled, and may not be useful in certain

situations. This was particularly emphasized in the context of patients who have a long history of suppression, patients who were not willing to discuss adherence, or for patients who either may not have the motivation to change their adherence patterns or had barriers to adherence that were not within their control such as unstable housing. As one provider recalled,

> "I can think of one patient [who] I've been repeatedly trying to get to start ART in the first place. . .[H]e doesn't have his own motivation for wanting to do it, which is the whole problem. . .and so in that case, [motivational interviewing] just doesn't work, because there is no motivation."
>
> –Provider, 6 years in HIV care

Providers in general felt that PSC was not an advantageous tool for assisting patients with adherence challenges compared to other counseling techniques. However, some advantages were still identified. Providers thought that PSC could be helpful for most of their patients who were already managing their health well. Additionally, providers felt that PSC could be useful in building trust and rapport between themselves and their patients. Lastly, many providers felt that because this counseling technique is patient-centered and elicits change and buy-in from the patient, it allows the patient to become the "main protagonist".

> "But I think, ideally, it is our role to help our patients come up with solutions, because I think most people like to come up with their own solutions. There's something about—interesting where the second technique [PSC] is more tied to listening, and kind of in a way making the patient the main protagonist, right?"
>
> –Provider, 10 years in HIV care

Providers expressed some concerns in utilizing PSC with their patients. One such concern surrounded limited knowledge on behalf of the providers themselves regarding the type of social services available to patients to assist with problem-solving. As one provider explained,

> "I would say that the challenges with this, is that the provider needs to educate themselves about all the different [resources] that we have to. . .reduce these barriers. . . And if you're a provider who is only passing through clinic or who's just going to shrug your shoulders and say, 'Oh, you should talk to the social worker about that'. That's going [make] that intervention. . .less effective."
>
> –Provider, 10 years in HIV care

Many providers also highlighted the fact that PSC success would be highly dependent upon the patient. For patients with adherence challenges or other social or mental health challenges, coming to a solution on their own may be particularly challenging. Providers believed that successful PSC requires a higher degree of health literacy than is common for patients with adherence challenges.

> "[W]hen I've used this approach before. . .some of these [challenges] are not easy to fix. Like, 'I'm homeless. I have no support in my life', you know, how do you say, 'Okay, let's come up with the solution for that'. That's not really practical in those cases."
>
> –Provider, 14 years in HIV care

**Arrange: Incorporating counseling into clinic practice.** During interviews, providers emphasized characteristics of the adherence intervention that would be required for successful integration into their practice. The intervention would need to provide very specific, non-judgmental language and phrases for prompts or scripts to ensure a level of consistency during implementation. Providers also stressed that, due to the limited amount of time allotted to each patient, the counseling component would need to be brief.

## Discussion

This formative work for the GAIN Study used the Five A's Framework to assess providers' perspectives on their current adherence counseling practices, the implementation of an ultra-brief adherence counseling intervention for use with their patients, and the feasibility and acceptability of incorporating POC NAT into that intervention (*see* Fig 1). Providers seemed most comfortable with MI and would prefer an adherence counseling intervention based in MI over PSC, although providers remained open to trying both counseling styles. The study noted general enthusiasm for and acceptability of POC NAT; however, providers expressed concerns with programmatic implementation of the proposed SAMBA II [18] primarily due to its two-hour wait time for results.

Though all providers explicitly stated that they would be open to learning either MI or PST, some speculated that other providers within their practice may be resistant to learning adherence counseling techniques. Although in one study high intensity MI training was shown to be effective in helping providers gain MI proficiency [28], other studies have had mixed success with provider adoption, highlighting individual characteristics [29] and the need for more in-depth participation outside of a workshop setting [30]. Additionally, ongoing peer and group supervision through continual training in MI or PST techniques has been shown to have wide success in development of competency and overall likelihood of successful integration into provider practice [31]. Our project might have greater success than past work, as there has been a shift in emphasis on the importance of changing the provider-patient dyad into a more collaborative framework to ensure behavioral changes [32, 33].

Providers interviewed suggested that the intervention package may not be applicable to all patients seen at Madison clinic and made suggestions as to how to target recruitment for the GAIN Study. These included having patients come in earlier to complete the POC NAT, altering clinic flow for patients who will see more than one provider or receive more than service and providing incentives such as coffee gift cards for patients to use while they wait for their results. Providers also felt there should be a focus on a subset of patients who are seen in the "Mod" clinic—who tended to be those most likely to be non-adherent to their ART. The population focus suggestion will be implemented when the GAIN Study begins recruitment.

POC NATs can reduce the time, cost, and resources it takes to transport, store, and process specimens for laboratory-based NATs by shifting the burden away from centralized laboratories, [27]. Evaluations of POC NATs outside the US have shown high sensitivity and specificity [28] and have shown to be potentially effective in ART monitoring [29] and could therefore be beneficial within the U.S. context. Among the providers in this study, the two-hour wait time for results was considered was considered the most significant barrier to implementation. The POC NAT being used in the GAIN Study was used successfully in Project DETECT, albeit in a non-implementation setting [17, 26]. As POC NATs are implemented in the clinic, work will be done to understand clinic flow and the average time spent in clinic and will attempt to address any concerns.

This study elicited perspectives from a range of providers who had experience with ART adherence challenges with persons living with HIV. The study benefitted from having the Five

A's, a strong conceptual model, which prospectively informed analysis. The Five A's framework has been used extensively within the U.S. for behavioral change for chronic conditions [23, 25] and has been highlighted as an integral component to future practitioner care [33].

This study had a few limitations. The small sample size of participating providers did not have direct experience delivering MI or PSC in conjunction with the POC NAT in a programmatic setting. Therefore, provider concerns about barriers to successful implementation are hypothesized, rather than experienced. Additionally, this was not a racially diverse population and providers were recruited from a single HIV clinic in Seattle, Washington, which limits the representativeness of these perspectives. Finally, implementation is planned with a specific POC NAT in mind, and the acceptability, feasibility, and concerns may differ if and when additional POC NATs become available in the U.S.

## Conclusion

The GAIN Study will be the first project to evaluate the implementation of POC NAT in the U.S. The proposed combined adherence counseling intervention and POC NAT were perceived by providers in this clinic to be acceptable. Continued formative work, including time-in-motion analyses of clinic flow, is ongoing and may show how best to address feasibility concerns about the two-hour time to results. In the next few years, the GAIN Study will pilot this intervention and may provide evidence to support the wide-scale adaptation of POC NAT to provide ART monitoring and promote ART adherence among PLWH. Future work with POC NATs could significantly improve the provision of HIV care and reduce morbidity and mortality associated with HIV infection.

## Supporting information

**S1 Checklist.**
(PDF)

## Acknowledgments

A special thank you to the Madison clinic providers, without whom this research would not have been possible, and the entire GAIN team for their support, patience and dedication.

## Author Contributions

**Conceptualization:** Deepa Rao, Joanne D. Stekler.

**Data curation:** Lauren Violette, Lisa Neimann.

**Formal analysis:** Dana L. Atkins, Lauren Violette, Lisa Neimann.

**Funding acquisition:** Joanne D. Stekler.

**Methodology:** Dana L. Atkins.

**Project administration:** Lisa Neimann.

**Supervision:** Deepa Rao, Joanne D. Stekler.

**Writing – original draft:** Dana L. Atkins.

**Writing – review & editing:** Lauren Violette, Lisa Neimann, Mary Tanner, Karen Hoover, Deepa Rao, Joanne D. Stekler.

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
