## [Decision Letter · Decision Letter 0]

12 Jan 2022

PONE-D-21-32892Provider Perspectives on the Use of Motivational Interviewing and Problem-Solving Counseling Paired with the Point-of-Care Nucleic Acid Test for HIV CarePLOS ONE

Dear Dr. Atkins,

Thank you for submitting your manuscript to PLOS ONE. After careful consideration, we feel that it has merit but does not fully meet PLOS ONE’s publication criteria as it currently stands. Therefore, we invite you to submit a revised version of the manuscript that addresses the points raised during the review process.

We look forward to receiving your revised manuscript.

Kind regards,

Bronwyn Myers

Academic Editor

PLOS ONE

Journal Requirements:

Reviewers' comments:

Reviewer's Responses to Questions

**Comments to the Author**

1. Is the manuscript technically sound, and do the data support the conclusions?

Reviewer #1: Yes

Reviewer #2: Yes

2. Has the statistical analysis been performed appropriately and rigorously? 

Reviewer #1: N/A

Reviewer #2: N/A

3. Have the authors made all data underlying the findings in their manuscript fully available?

Reviewer #1: Yes

Reviewer #2: No

4. Is the manuscript presented in an intelligible fashion and written in standard English?

Reviewer #1: Yes

Reviewer #2: Yes

5. Review Comments to the Author

Reviewer #1: Thank you for the opportunity to review this manuscript. It addresses an important context. I provide recommendations for improvement that could strengthen its relevance and impact

Introduction:

Paragraph 2/3/4. PSC or problem solving therapy- also successfully used to address alcohol use and HIV adherence. Apart from the STRIVE trial, there are two trials on MI-PST among PLWH in South Africa that have just been completed and that show MI-PST is feasible to deliver, acceptable to patients and providers and has promising outcomes. I’ve added references to the trial protocols and formative/process evaluation work here.

Myers, B., Parry, CDH, Morojele, N., Nkosi, S., Shuper, P., Kekwaletswe, C., Sorsdahl, K. (2020). “Moving forward with life’: Acceptability of a brief alcohol reduction intervention for people receiving antiretroviral therapy in South Africa. International Journal of Environmental Research and Public Health. 17(16):5706. doi: 10.3390/ijerph17165706.

Myers, B., Lund, C., Lombard, C., Joska, J., Levitt, N., Butler, C., Cleary, S., Naledi, T., Milligan, P., Stein, D., Sorsdahl, K. (2018). Comparing dedicated and designated models of integrating mental health into chronic disease care: study protocol for a cluster randomized controlled trial. BMC Trials, 19: 185. DOI: 10.1186/s13063-018-2568-9.

Parry, C.D., Morojele, N.K., Myers, B.J., Kekwaletswe, C.T., Manda, S.O.M., Sorsdahl, K., Ramjee, G., Hahn, J.A., Rehm, J., & Shuper, P.A. (2014). Efficacy of an alcohol-focused intervention for improving adherence to antiretroviral therapy (ART) and HIV treatment outcomes - a randomised controlled trial protocol. BMC Infectious Diseases, 14: 500. doi:10.1186/1471-2334-14-500.

Petersen Williams, P., Brooke-Sumner, C., Joska, J., Kruger, J., Vanleeuw, L., Dada, S., Sorsdahl, K., Myers, B. (2020). Young South African Women on Antiretroviral Therapy Perceptions of a Psychological Counselling Program to Reduce Heavy Drinking and Depression. Int. J. Environ. Res. Public Health, 17, 2249. https://doi.org/10.3390/ijerph17072249.

For the third paragraph- perhaps given an example or two of non-motivational factors that impact on adherence- I would suggest highlighting mental health and substance use concerns, especially as the PST literature is very much focused on the benefits of helping people develop adaptive problem solving and coping skills to manage negative emotions and avoid substance use coping.

Please can you revise the final paragraph of the introduction so that the rationale for conducting this study, the contribution to the literature and the specific aims of this paper are clear.

Materials and methods: Overall the methods needs to be fleshed out to comply with standard reporting guidelines for qualitative research

1. Please ensure that you follow the COREQ or similar reporting guidelines for qualitative research and include the appropriate checklist as a supplementary file.

2. Under study design and population, in keeping with COREQ guidelines, a description of the design needs to be explicitly made. Here it would be appropriate to note that the formative work is embedded in a broader study. In terms of setting, it would be useful to describe and explain the kinds of clinics offered- ie what is a moderate needs clinic and how does it differ from other levels of need?

3. Who developed the IDI guide and was it piloted?

4. Please explain the process of recruiting providers, obtaining their consent to participate. How were they approached and by whom? Did anybody refuse to participate and if so why?

5. More information is needed about the qualifications and background of the staff who conducted the interviews as per COREQ guidelines

6. I am struggling to see the relevance of the 5 a’s framework for HIV counselling- please can you provide more information on why this and not another framework was chosen. At a minimum this framework needs to be described much more fully, possibly in the background as a a way of trying to understand how MI-PST approaches can be integrated into the various components of counselling,

7. Was there any member checking of the themes?

Results

Please provide a summary of the main themes that emerged from the IDIs and how these relate to each other.

Reading through the results, I think there is an opportunity to present these more critically. For instance, uin the first theme, it is clear that psychosocial problems impact on adherence, but in the second theme, the providers are not advising patients on how to manage these challenges, but are focused on “telling” patients to be adherent. So it is noteworthy that they do not currently use a patient-centred approach. A lot of the results can be streamlined and stated more succinctly so the narrative flows through the themes.

Discussion. I have few comments here, except for a reflection on the importance of supervision and support when training providers in MI-PST techniques. Ongoing peer and group supervision can elp providers develop competency and confidence in their practice beyond a single workshop and increases the likelihood that they will use the skills. It is also a vehicle for brainstorming challanges/resources with colleagues.

This article reflects on this:

Jacobs, Y., Myers, B., van der Westhuizen, C. et al. Task Sharing or Task Dumping: Counsellors Experiences of Delivering a Psychosocial Intervention for Mental Health Problems in South Africa. Community Ment Health J 57, 1082–1093 (2021). https://doi.org/10.1007/s10597-020-00734-0

Reviewer #2: 1. In the first paragraph, the last line of the of the introduction section, it states: “One strategy for adherence monitoring and support is provider counseling”. What the authors mean by “provider counseling”? It is counseling for providers? Or counseling by providers? In both cases I couldn’t attend well.

Methods

Dear Editor, thank you for inviting this manuscript. It is important in that it is combined with PSC to improve adherence.

1. “Following consensus coding, the remaining nine transcripts were independently coded by one member of the coding team. All transcripts were then coded by another member of the team, and disagreements in code application were noted.” In the first sentence, I understand that the coding of the remaining transcripts was done by one of the member of the team. In the second sentence, I could understand that it was done by another member of the team. Would the authors clarify this?

2. I was looking for the description about the MI and the PSC in the methods though I couldn’t. how the authors implemented the MI and the PSC? For how many sessions? At what setting? For how long? This should have been stated so that the readers can judge about the basics of the interventions. If the authors have another paper in this aspect, it should be cited and linked.

Results

3. In the main results section, it looks that authors reported the findings using thematic topics: ex: Current Assessment and Advising Practices, Creation of the GAIN model, Evaluating the Point-of-Care NAT Intervention Using the Five A’s Framework, Assess: Using the Point-of-Care NAT for Adherence Assessment, Arrange: Incorporating POC NAT into Clinic Flow, etc. However, the results section of the abstract does not contain at least some of these topics.

4. What is “SAMBA II”?

6. PLOS authors have the option to publish the peer review history of their article (what does this mean?). If published, this will include your full peer review and any attached files.

Reviewer #1: No

Reviewer #2: No

---

## [Author Response · Author response to Decision Letter 0]

12 May 2022

Thank you for your time in reviewing this manuscript. I have attached to the submission the responses to your comments. Please let me know if anything else is needed.

---

## [Decision Letter · Decision Letter 1]

8 Jun 2022

Provider Perspectives on the Use of Motivational Interviewing and Problem-Solving Counseling Paired with the Point-of-Care Nucleic Acid Test for HIV Care

PONE-D-21-32892R1

Dear Dr. Stekler,

We’re pleased to inform you that your manuscript has been judged scientifically suitable for publication and will be formally accepted for publication once it meets all outstanding technical requirements.

Kind regards,

Bronwyn Myers

Academic Editor

PLOS ONE

Additional Editor Comments (optional):

Reviewers' comments:

Reviewer's Responses to Questions

**Comments to the Author**

1. If the authors have adequately addressed your comments raised in a previous round of review and you feel that this manuscript is now acceptable for publication, you may indicate that here to bypass the “Comments to the Author” section, enter your conflict of interest statement in the “Confidential to Editor” section, and submit your "Accept" recommendation.

Reviewer #1: All comments have been addressed

Reviewer #2: All comments have been addressed

2. Is the manuscript technically sound, and do the data support the conclusions?

Reviewer #1: (No Response)

Reviewer #2: Yes

3. Has the statistical analysis been performed appropriately and rigorously? 

Reviewer #1: (No Response)

Reviewer #2: N/A

4. Have the authors made all data underlying the findings in their manuscript fully available?

Reviewer #1: (No Response)

Reviewer #2: Yes

5. Is the manuscript presented in an intelligible fashion and written in standard English?

Reviewer #1: (No Response)

Reviewer #2: Yes

6. Review Comments to the Author

Reviewer #1: (No Response)

Reviewer #2: (No Response)

7. PLOS authors have the option to publish the peer review history of their article (what does this mean?). If published, this will include your full peer review and any attached files.

Reviewer #1: No

Reviewer #2: No

---

## [Editor Report · Acceptance letter]

13 Jun 2022

PONE-D-21-32892R1 

Provider Perspectives on the Use of Motivational Interviewing and Problem-Solving Counseling Paired with the Point-of-Care Nucleic Acid Test for HIV Care 

Dear Dr. Stekler:

I'm pleased to inform you that your manuscript has been deemed suitable for publication in PLOS ONE. Congratulations! Your manuscript is now with our production department. 

Kind regards, 

on behalf of

Dr. Bronwyn Myers 

Academic Editor

PLOS ONE